# A New Machine Learning Forecasting Algorithm Based on Bivariate Copula Functions

**J. A. Carrillo** [1], **M. Nieto** [1], **J. F. Velez** [2] **and D. Velez** [1,*]

[1] Department of Statistics and Operational Research, Faculty of Mathematics, Complutense University of Madrid, 28040 Madrid, Spain; juan.carrillo.segura@alumnos.upm.es (J.A.C.); miriamnieto@ucm.es (M.N.)

[2] Department of Computer Science, Escuela Tecnica Superior de Ingenieria Informatica, Universidad Rey Juan Carlos, Mostoles, 28933 Madrid, Spain; jose.velez@urjc.es

[*] Correspondence: danielvelezserrano@mat.ucm.es

**Abstract:** A novel forecasting method based on copula functions is proposed. It consists of an iterative algorithm in which a dependent variable is decomposed as a sum of error terms, where each one of them is estimated identifying the input variable which best "copulate" with it. The method has been tested over popular reference datasets, achieving competitive results in comparison with other well-known machine learning techniques.

**Keywords:** copula; machine learning; conditional probabilistic forecast

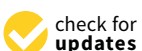



## 1. Introduction

One way to make predictions of a variable, $Y$, from the values of another, $X$, is to compute the conditional expectation of $Y|X$. In the continuous case for example, if the $g(y|x)$ function defines its conditional density, this expected value is calculated according to Equation (1), with $D_Y$ being the support of the $Y$ variable.

$$E[Y \mid X = x] = \int_{D_Y} y g(y \mid x) dy \tag{1}$$

In this equation, the conditional density function $g(y|x)$ can be computed from the joint density function of $(X, Y)$, $h_{XY}(x, y)$, and the marginal one of $X$, $f_X(x)$, using Equation (2).

$$g(y|x) = \frac{h_{XY}(x, y)}{f_X(x)} \tag{2}$$

Estimating marginal functions can be reasonably easy using estimators such as histograms, frequency polygons, Na-daraya-Watson kernels, or empirical distribution functions.

However, it is more complicated to propose a regular expression for joint distributions. One of the reasons that justifies this difficulty is the need to represent the true implicit dependence relationship between the variables.

In fact, if $F_X$ and $G_Y$ are the marginal cumulative distribution functions associated with $X$ and $Y$, respectively, then it can be demonstrated that there are infinity joint cumulative distribution functions $H_{X,Y}$ with these marginals [1]. Furthermore, pairs of variables can be found, $(X_1, Y_1)$ and $(X_2, Y_2)$, where both $X_j$ and $Y_j$ variables have the same distribution with the same linear $\rho_{X_j, Y_j}$ correlation coefficient too, but with different dependence structures—that is, there may be two different joint distribution functions, $H_1$ and $H_2$, associated with $(X, Y)$, which would explain the dependency relationship between $X$ and $Y$ in a different way.

For that reason, it is important to identify the joint cumulative distribution function $H_{XY}$ that truly reflects the relationship between $X$ and $Y$. Later, the conditional distribution could be built from it to make predictions using Equation (1). The methodology proposed in this paper is based on the estimation of these distributions through copula functions.

There are few works in which copulas are linked to the concept of machine learning. Certainly, when searching for "TS = (Copula* NEAR ('data mining' OR 'machine learning'))" in Web of Science (apps.webofknowledge.com, accessed on 25 May 2021), only 15 references were found. In the survey "Copulas in machine learning" [2] (2013), the author talks about the need for synergy between machine learning and copula frameworks and implores the researchers in the copula community to develop algorithms able to cope with high-dimensional challenges. To cover this shortcoming, a novel forecasting methodology that uses bivariate copulas is proposed. This methodology is an extension of a thesis published in 2007 [3], in which copula functions were used to forecast, from incremental temperature variables, the percentage error term obtained after adjusting an ARIMA model to a gas demand series. The methodology consisted of an iterative process in which copula functions were used to relate the residual processes with the input variables, as an alternative to the traditional way in which these input variables are included in the ARIMA equation (ARIMAX model).

The work presented in this paper is a generalization of this methodology, incorporating some of the popular ideas of machine learning methods. For example, there is a selection process in which error terms are linearly (in an additive sense) and sequentially predicted through copula functions, identifying the best pair (input variable, copula) for this end. It remembers the stepwise selection process, sometimes used in adjusting regression model, but with the difference that each variable can participate several times in a similar way that they can do in a random forest or a gradient boosting model. During this iterative process, training and validation datasets are distinguished, using an *early stop criterion* frequently used in well-known methods such as neural networks or the mentioned trees ensemble methods. The methodology has been tested over several reference datasets unlike the most competitive machine learning methods, achieving better results in some of them.

The paper is structured as follows: Section 1.1 consists of a revision of the state of the art about the use of copulas for making predictions; in Section 2, the proposed forecasting methodology based on this kind of functions is introduced; in Section 3, this methodology is applied to several reference datasets, comparing the corresponding results with the achieved ones by the most competitive machine learning techniques; finally, in Section 5, the conclusions and the future lines of work are presented.

### 1.1. State of the Art

Nowadays, there are many fields in which copula functions are used to model multivariate relationships. They are usually associated mainly with the Economy,but their use has spread to various sectors, finding multiple applications in financial [4], insurance [5], energy [6], meteorological [7] and, more recently, forestry and environmental sciences [8].

From a mathematical point of view, copulas are frequently associated with simulation [9]. In this context, copulas are frequently used in risk management [10], where they allow simulating future scenarios taking into account the financial structure of the market. So, S. Ortobelli et al. [11] compared the performance of several reward risk strategies based either on simulated data through copula functions or on historical ones, demonstrating the better performance of strategies valued on the first ones. Copulas have been used too when available data are scarce and they are insufficient to quantify possible risks associated with especially adversarial events, allowing the simulation of samples of pairs of copulated outliers. For example, in the work of R. De Matteis [12], extreme-value copulas [13] were used to simulate the simultaneous occurrence of highly expensive building and contents accidents in an insurance company, and in the work of C. P.Khedun et al. [14], they are used to simulate precipitation anomalies. In the context of outliers, there are some references associated with the use of copulas for the detection of this type of data. For example, T. Bellini [15] uses elliptical copulas to detect multivariate atypical observations, and K. Domino [16] uses the t-Student copulas to artificially sample outliers.

Moving between the fields of simulation and prediction, copula functions are frequently used in time series analysis, finding an interesting review of copula-based models

for economic and financial time series data in the work of A. Patton [17]. These series are usually dominated by a random walk component [18], as the aim is frequently focused on forecasting not its average value but its variance. In this context, there are some interesting contributions from copula modelling. O. Sokolinskiy et al. [19] forecast volatility one-day ahead for a financial index, finding that the copula-based realized volatility model (C-RV) outperforms conventional forecasting in terms of accuracy and efficiency. A. Kresta [20] analyzes the applicability of the copula-GARCH model in portfolio optimization, simulating the evolution of financial time series and demonstrating that they provide better forecasts than a benchmark based on bootstrapping techniques. There are also studies that make interesting comparisons between the accuracy of the copula-GARCH and Dynamic Conditional Correlation (DCC) models for forecasting the Value-at-Risk (VaR) and expected shortfall of bivariate portfolios [21,22]. Regarding the VaR metric, S. Guharay [23] proposed a more robust estimation of it based on copula functions. Other authors make comparisons between the well known Capital Asset Pricing Model (CAPM) and copula functions to analyze the co-movement and dependence structure between indexes. So, R. Luo and M. Bhatti [24] use Gaussian, symmetrized Joe-Clayton, and Rotated Gumbel Copulas to conduct this kind of analysis on Islamic investment fund data, while F. Mansor et al. [25] proposed CAPM as an alternative to complex copulas and DCC models. Finally, within this field of capturing co-movements between time series associated with indexes, C. Nguyen [26] proposed a new class of mixed copulas from Clayton, Joe, Gumbel, and Joe copulas, achieving interesting results for investors to configure its portfolios. Di Clemente [27] proposes an interesting methodology for measuring and optimizing the credit risk of a portfolio following a copula-based approach.

Finally, within the scope of forecasting, but outside the one of time series, it is well known that the use of vine copulas [28,29], based on the theory introduced by H. Joe [30] and T. Dedford and R. Cooke [31]. These copulas allow the construction of multivariate distributions from simple building blocks called pair-copulae, decomposing a multivariate non-Gaussian distribution into a product of conditional and unconditional distribution functions, not requiring natural conditional independence assumptions. Certainly, some authors [29] solve prediction problems using copulas, showing that algorithms based on them achieve better results than linear regression models. However, to the extent of the knowledge of the authors of this paper, these algorithms are not usually compared with the provided ones by other popular machine learning methods such as, for example, neural networks, random forest, or gradient boosting.

As some of the cited works make a comparison between copula and DCC, CAPM or regression models, in the same way, in the present work, a comparison between the aforementioned machine learning techniques and the *Additive Decomposition Algorithm Based On Copulas* (ADABOC) is carried out. This comparison demonstrates that our proposal achieves competitive results.

## 2. Materials and Methods

This chapter details the methodology proposed for predicting an interval variable using copula functions. In Section 2.1, a brief introduction to these functions will be presented to facilitate the understanding of the mentioned methodology, which is detailed in Section 2.2.

### 2.1. Copula Functions: Preliminary Concepts and Results

The word copula was originally used in a statistical context in 1959 by the mathematician Abe Sklar in a theorem which bears his name [32]. With this term, the author referred to functions that join (or copulate) multivariate distribution functions to their unidimensional marginals [33].

Essentially, a copula is a multivariate cumulative distribution whose marginal functions are distributed according to standard uniforms. The exposition will be focused on the

bivariate case because, apart from being the most studied and referenced in the literature, it is the needed one to understand the proposed algorithm.

**Definition 1** (Bicopula function). *A bicopula is a function $C : [0,1]x[0,1] \rightarrow [0,1]$ with the following properties:*

1.  $C(u,0) = 0; C(0,v) = 0;$
2.  $C(u,1) = u; C(1,v) = v;$
3.  *C is non-decreasing: for each hyperrectangle $B = [u_1, u_2]x[v_1, v_2]$, its volume is non-negative:* $V_C(B) = C(u_2, v_2) - C(u_2, v_1) - C(u_1, v_2) + C(u_1, v_1) \geq 0.$

Sklar's theorem is the main result of the copula theory, since it establishes the relationship between the joint distribution and univariate marginals through this type of function.

**Theorem 1** (Sklar Theorem). *Let X and Y be random variables with marginal distribution functions $F_X$ and $G_Y$, respectively, and a joint distribution function $H_{XY}$. Then, there exists a copula C such that:*

$$H_{XY}(x,y) = C(F_X(x), G_Y(y)) \quad \forall x, y \epsilon \bar{R} = [-\infty, \infty]. \tag{3}$$

*If $F_X$ and $G_Y$ are continuous, then C is unique. If not, C is uniquely determined in $Range(F_X) \times Range(G_Y)$. Reciprocally, if C is a copula and $F_X$ and $G_Y$ are distribution functions, then the function $H_{XY}(x,y) = C(F_X(x), G_Y(y))$ is a joint distribution function with marginals $F_X$ and $G_Y$.*

It is important to remark on the observation referring to the non-continuous case, as most applied papers in the literature that use copula models that deal with continuous data. Certainly, [34,35] talks about whether it should be advised to make inferences in copula models with discrete data. The author of the first one argues that modeling a discrete distribution with a parametric copula and its marginal functions can be very difficult. This is the reason that the continuous case will be considered.

On the other hand, note that, if X is a random variable with a distribution function $F_X(x)$ ($F_X(x) = P[X \leq x]$), then $F_X(X)$ can be seen as a random variable too. As a result, if $F_X$ is invertible, then:

$$P[F_X(X) \leq x] = P[X \leq F_X^{-1}(x)] = F_X(F_X^{-1}(x)) = x \quad \forall x \epsilon D_X \tag{4}$$

Observe that this is the definition of a standard uniform distribution. For this reason, $F(X)$ (and $G(Y)$), can be considered as an uniform variable $U$ ($V$, respectively). So, Equation (3) can be rewritten as:

$$H_{XY}(x,y) = C(u,v) \tag{5}$$

In other words, Sklar's theorem allows us to estimate the joint distribution function $H_{XY}$, finding the copula function C, which better fits to the values $(u, v)$ of the uniform variables $(F_X, G_Y) = (U, V)$. The relevance of this result lies in the fact that a lot of copula functions which reflect a different types of relationships have been studied, making the identification of a good estimation for $H_{XY}$ easier.

Sklar's theorem can be used to identify the independence case between variables too. In fact, when the copula function that best fits the uniform pairs $(u, v)$ is the product copula (6), X and Y can be considered as independent variables.

$$\Pi(u,v) = u * v \tag{6}$$

This is easy to demonstrate:

$$H_{XY}(x,y) = C(F_X(x), G_Y(y)) = \Pi(u,v) = u \cdot v = F_X(x) \cdot G_Y(y). \tag{7}$$

It has already been emphasized that we need to know the conditioned distribution function $G_{Y|X}$ to make predictions of the $Y$ variable from the one $X$. As $G_{Y|X}$ distribution is difficult to be directly estimated, it seems reasonable to use copulas to this end, as it is important to understand the role these functions play in the characterization of $G_{Y|X}$. For this, conditioned copulas associated with a copula, C, will be introduced.

**Definition 2** (Conditioned copulas associated with a copula). *Let C be a copula function.*

*Fixed $U = u$, copula conditioned to u is a function of V variable:* $C_1(u,v) = C(v|u) = \frac{dC(u,v)}{du}$

*Fixed $V = v$ copula conditioned to v is a function of U variable:* $C_2(u,v) = C(u|v) = \frac{dC(u,v)}{dv}$

It can be demonstrated (see Theorem 2.2.7 in [33]) that these partial derivatives exist almost certainly for all u and v, except Lebesgue null measurement sets, and that they almost certainly do not decrease in the unit interval.

So, as a consequence of the adaptation of Sklar's theorem to continuous conditioned distributions, the next result is presented [36].

**Proposition 1** (Conditioning with copulas). *Let C be a copula function and $C_1(u,v)$ be the derivative of $C(u,v)$ with respect to U. If the joint distribution of X and Y is given by $H_{XY}(x,y) = C(F_X(x), G_Y(y))$, then the conditional distribution of $Y|X = x$ is given by:*

$$G_{Y|X}(y) = C_1(F_X(x), G_Y(y)) \tag{8}$$

Observe again that in the independent case $(C(u,v) = u \cdot v)$, the conditioned distribution of $V$ given $U = u$ is $C_1(u,v) = \frac{dC(u,v)}{du} = \frac{d(u \cdot v)}{du} = v$, which is independent of the value of U.

### 2.2. Prediction Algorithm Based on Bivariate Copula Functions

In this subsection, a method for adjusting supervised models to interval variables through bivariate copula functions is proposed. It is important to note that the term "copula" will be used even when the algorithm actually refers to bicopulas, meaning the bidimensional version of this function.

The adjustment process consists of an iterative algorithm that refines a starting basic predictor defined by the average of the values of the dependent variable. This one generates an initial error term. In the first step, the aim is to identify the variable which better explains it through a copula function. The selection of the most appropriated copula to this end is another task to solve in the algorithm. Then, as a result of applying the pair formed by the input variable and the copula function to explain the first error term, a new one is generated, which leads to the start point of step two. This process is repeated until a stopping criterion is satisfied.

In summary, three phases were sequentially applied in each one of the steps of the iterative algorithm:

1. Adjustment phase: selection of the copula function $C^*$, which best models the relationship between explanatory variables and the residual term obtained in the previous step.
2. Prediction phase: use of the conditioned copula $C_1^*$ to estimate the value of the error term derived from the previous predictor, which is updated by adding up the mentioned estimated value.
3. Assessment phase: Evaluation of the goodness of the new predictor over a validation dataset. Mean Absolute Error (MAE) has been the metric used to this end in the proofs presented in Section 3.

On the other hand, three are three possible criteria that can stop the algorithm:

1.  Independence criterion: the best copula $C^*$ selected is the product one, indicating the independence between explanatory variables and the error term to be predicted.
2.  Early stopping criterion: the evaluated metric has not significantly improved during the last steps.
3.  Maximum iterations criterion: the maximum number of iterations prefixed is reached.

The methodology of the proposed algorithm, schematically illustrated in Figure 1, is common to any supervising model to build a predictor $\hat{Y}$ of the dependent variable $Y$ that optimizes the value of a metric previously established. The fact that each predictor could be generated from simulated values is an advantage. Apart from averaging them to obtain the usual mathematical expectation, it is possible to construct a more robust predictor, such as the median, to avoid the effect of possible outliers. From here, the mean predictor will be considered so as not to favor the results provided by the algorithm proposed against other machine learning techniques tested in Section 3, severely affected by the effect of outliers (such as the GLM model).

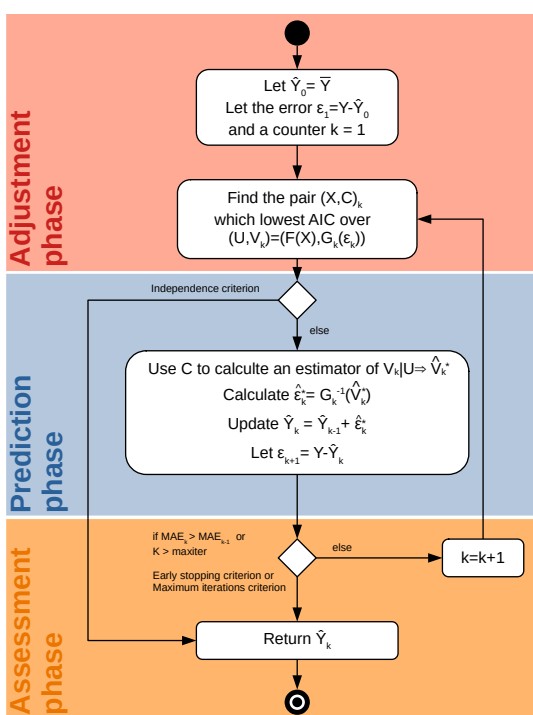

**Figure 1.** Copula selection process.

In any case, the predictor is constructed from the input variables $\{X_1, X_2, \ldots, X_m\}$, with $\{y(i), x_1(i), x_2(i), \ldots, x_m(i)\} \, \forall i \in \{1, 2, \ldots, n\}$ being a sample of values corresponding to them. Uppercase and lowercase letters are used to refer to random variables and observed data, respectively. The value in brackets *i* refers to each one of the *n* observations.

In the adjustment phase, three different datasets or samples of observations were used: one for training, another one for validation, and the third one for the test, with sizes of $n_1$, $n_2$ and $n_3$, respectively. The training sample was used to decide, step by step, the input variable and the copula functions involved in the construction of the predictors. These were evaluated, respect to the prefixed metric, over the validation dataset. The aim of the latter step is to favor the generalization capacity of the predictor, which will be finally tested using the test dataset. As has been previously mentioned, the MAE will be considered as the metric to be minimized:

$$MAE = \frac{1}{n_2} \sum_{i=1}^{n_2} |Y(i) - \hat{Y}(i)| \tag{9}$$

Note that this metric is averaged over the $n_2$ observations associated with the validation table.

In addition to the use of a validation table, there are two additional parameters taken into account to avoid possible overfitting effects. Both of them also allowed us to reduce the computational time cost of the algorithm:

- On the one hand, an early stopping criterion [37] has been established. According to this criterion, the adjustment process stops if the assessment of the metric over the validation sample does not improve their values after several iterations initially prefixed by the *earlyStoppingInterations* parameter.
- On the other hand, different subsamples of the training dataset are considered in each one of the iterations of the process. The size of these subsamples is specified through the *subsamplePercent* parameter, which specifies the percentage of the $n_1$ training dataset observations randomly selected in each iteration. This modeling strategy is, for example, used on stochastic gradient boosting algorithm [38] as an improvement of the traditional gradient boosting method [39]. The influence of this parameter is tested in Section 3.2.

Next, the tasks and calculus involved in each one of the steps of the algorithm will be detailed.

### ADABOC Algorithm

To start with, a beginning predictor, given by the average value of $Y$, is considered. This mean value is calculated with the $n_1$ observations conforming to the training dataset:

$$\hat{Y}_{(k=0)} = \hat{\beta}_0 = \bar{Y} = \frac{1}{n_1} \sum_{i=1}^{n_1} y(i) \tag{10}$$

This predictor generates a first variable associated with the corresponding error term:

$$\varepsilon_{(k=1)} = Y - \hat{Y}_{(k=0)} \tag{11}$$

and through it, a first MAE is obtained, which can be evaluated in the three datasets defined above:

$$MAE_{(k=1)} = \frac{1}{n_d} \sum_{i=1}^{n_d} |y(i) - \hat{y}_{(k=0)}(i)| \quad \forall d \in \{1, 2, 3\}. \tag{12}$$

The idea is to refine throughout a maximum of iterations, specified by the *maxiter* parameter, the predictor initialized in Equation (10). This refinement process consists of identifying, in each iteration $k$, the copula function that better links the sequentially obtained variables $\varepsilon_k$ with some of the explanatory ones $X_j$. To this end, a set of copula function families to be tested must be previously established: $\{C^1, C^2, \ldots, C^p\}$.

To ensure both $X_j$ and $\varepsilon_k$ are in the definition domain of the copula functions (the unit rectangle $[0,1] \times [0,1]$), these variables must be transformed through their cumulative distribution functions, $F_{X_j}$ and $G_{\varepsilon_k}$, respectively. It can be observed that only G depends on the subscript $k$. This is because the variable $X_j$, and therefore the function $F_{X_j}$, will not change from one iteration to another. In contrast, $\varepsilon_k$ represents the dependent variable to be predicted in the $k^{th}$ step, which has been modified according to the predictor constructed in the previous one (see Equations (10) and (11)). For that reason, the subscript $j$ represents the independent variables ($X_j$ or the transformation $F_{X_j}(X_j) = U_j$), while the subscript $k$ will refer to the residuals obtained by predicting the dependent variable ($\varepsilon_k$ or the transformed one $G_{\varepsilon_k}(\varepsilon_k) = V_k$). Note that these are the only terms that will be modified in each iteration.

Let $\{u_1(i), u_2(i), \ldots, u_m(i), v_k(i)\}$ be the uniform values obtained from the data $\{x_1(i), x_2(i), \ldots x_m(i), \hat{\varepsilon}_k(i)\}$ through the marginal functions $F_{X_1}, F_{X_2}, \ldots, F_{X_m}, G_{\varepsilon_k}$ associated with $X_1, X_2, \ldots, X_m, \varepsilon_k$ variables, respectively. Observe that a *hat* symbol is used to distinguish

between the error variable $\varepsilon_k$ and its estimator $\hat{\varepsilon}_k$. The corresponding estimated values for the latter, calculated in each one of the iterations, will be referred to as $\hat{\varepsilon}_k(i)$.

The task of identifying the marginal functions can be completed by trying to find the closer theoretical distribution from a list of the most common ones. This way, for each random variable, it would be necessary to propose hypothesis tests associated with popular distributions and select the one that provides a lower *p*-value. However, this could be inadequate in case none of them would adjust well enough. Alternatively, kernel estimators have been used in the proposed method. Other authors (see [12]) use the continuous version of the empirical distribution function instead.

Once the uniform sample has been generated, the following task consists of identifying, for each one of the $U_j$ variables (associated with each one of the $X_j$ input variables), the copula function that best fits the pairs $(u_j(i), v_k(i))$. This task is carried out in two stages:

- Firstly, one copula associated with each one of the *p* families is selected. The selection process consists of estimating the value of the parameters that the family depends on. The estimation is carried out using the training dataset, applying the maximum likelihood method as proposed in [12]. Proceeding this way, a total of $m \cdot p$ copula functions, $C_{j,k}^r$, is obtained, one per input variable and copula family.
- Secondly, representative values of the fitting goodness of these $m \cdot p$ functions to the pairs $(u_j(i), v_k(i))$ are calculated. To this end, some metrics can be acquired, such as the value of the likelihood function, the Akaike information (AIC), or the Bayesian inference criterion (BIC). The first metric has been used due to several authors [12,40,41] considering it as a good criterion.

Once these $m \cdot p$ values have been calculated (see Table 1), the smallest of them is considered representative of the best fitting to the training data $(u_j(i), v_k(i))$. Let $(X_j^*, C_{j,k}^*)$ be the pair chosen as the optimal one at the end of the $k^{th}$ step. To simplify the notation in the algorithm, $C_k^*$ will be used, occasionally, to denote the copula function chosen in this iteration:

$$C_k^* = \underset{j,r}{\operatorname{argmin}}\{AIC(C_{j,k}^r)\} \tag{13}$$

**Table 1.** AIC calculated by pair (variable/copula).

| Variable\Copula | $C^1$ | $C^2$ | ... | $C^p$ |
|---|---|---|---|---|
| $X_1$ | $AIC_{1,k}^1$ | $AIC_{1,k}^2$ | ... | $AIC_{1,k}^p$ |
| $X_2$ | $AIC_{2,k}^1$ | $AIC_{2,k}^2$ | ... | $AIC_{2,k}^p$ |
| ... | ... | ... | ... | ... |
| $X_m$ | $AIC_{m,k}^1$ | $AIC_{m,k}^2$ | ... | $AIC_{m,k}^p$ |

At this point, the independence stopping criterion is tested. This way, in the case in which $C_k^* = \Pi$, there would be not able to explain the error term through a copula function. As a result, independence between all the variables $X_j$ and $\varepsilon_k$ is concluded and the algorithm stops. Otherwise, a predictor of $\varepsilon_k$ conditioned to the values of $X_j^*$ is generated:

$$\hat{\varepsilon}_{(k=1)} = E[\varepsilon_{(k=1)}|X_j^* = x_j] \tag{14}$$

According to this expression, the expected values $\hat{\varepsilon}_{(k=1)}(i)$ are estimated for each value $x_j(i)$ of the variable $X_j^*$.

This predictor allows one to obtain a new predictor for the dependent variable *Y*:

$$\hat{Y}_{(k=1)} = \hat{Y}_{(k=0)} + \hat{\varepsilon}_{(k=1)} = \bar{Y} + \hat{\varepsilon}_{(k=1)} \tag{15}$$

Observe that the conditioned mathematical expectation of Equation (14) must be calculated using the density $g_{\varepsilon_k|X_j^*}$, which is unknown. However, the already estimated function $C_k^*$ contains information about the dependency relationship between $X_j^*$ and $\varepsilon_k$

through the transformed variables $U_j = F_{X_j^*}(X_j^*)$ and $V_k = G_{\varepsilon_k}(\varepsilon_k)$. Hence, it can be used to estimate $g_{\varepsilon_k|X_j^*}$. Specifically, it consists of simulate values of the variable $V_k|U_j = u_j$, being $u_j = F_{X_j^*}(x_j^*)$, detransform them using $g_{\varepsilon_{(k=1)}}^{-1}$ to obtain the corresponding values of the variable $\varepsilon_{(k=1)}$ and finally average the latter.

The simulation of the values of the variable $V_k|U_j = u_j$ can be carried out using the inverse transform method [42]. This method requires that the function $C_{j,k|1}^*$ (or $C_{k|1}^*$ to simplify) admits an explicit expression. This function is the conditioned copula derived from the known copula $C_k^*$, in which the subscript "1" is used to refer to the conditioning with respect to the first of the variables $(U_j)$ of the pair $(U_j, V_k)$. However, there are copula functions for which this explicit expression does not exist and, as a result, the corresponding inverse function can not be expressed in a closed-form [12] to apply the mentioned method.

Alternatively, an approximation is proposed for estimating Equation (14). The question is how to weigh the $g_{\varepsilon_{(k=1)}}^{-1}(v)$ values to average them or, in other words, how many values must be generated for each $v \in [0, 1]$, which is the support of the variable V. To answer this question, note that the density function $c_{j,k}^*$ (or $c_k^*$ to simplify) associated with the known copula $C_k^*$, contains information about the proportionality relationship that must exist between the $v$ values to be generated, and so, Z it can be used to assign the mentioned weights. According to this consideration, the proposed method consists of:

- Let $\{v_1, v_2, ..., v_t\}$ be equidistant values $\in [0, 1]$ and $\hat{\varepsilon}_{(k=1),s} = g_{\varepsilon_{(k=1)}}^{-1}(v_s)\forall s \in \{1, 2, ..., t\}$. The number of values is specified by the parameter *numBins* (see Algorithm 1).
- Let $w_s = c_{k|1}^*(v_s|u_j) = \frac{c_k^*(u_j, v_s)}{f_{U_j}(u_j)}$ be the weight associated with each $\hat{\varepsilon}_{(k=1),s}$ value $\forall s \in \{1, 2, ..., t\}$.
  Note that the value $c_k^*(u_j, v_k)$ can be calculated because $c_k^*$ is known and has an explicit form. On the other hand, $f_{U_j}(u_j) = \int_{[0,1]} c_k^*(u_j, v)dv$ is an area that can be easily estimated using numerical methods. Again, equidistant points are generated for approximating these areas using the *numBins* parameter.
- Let $\hat{\varepsilon}_{(k=1)} = \sum_{s=1}^{t} w_s \cdot \hat{\varepsilon}_{(k=1),s}$.
- Construct the predictor $\hat{Y}_{(k=1)} = \bar{Y} + \hat{\varepsilon}_{(k=1)}$.

The variable corresponding to the error associated with the latter predictor of the dependent variable $Y$ is:

$$\varepsilon_{(k=2)} = Y - \hat{Y}_{(k=1)} \tag{16}$$

Figure 2 summarizes the construction of the first predictor $\hat{Y}_{(k=1)}$ from original data values $(X_j, Y)$. $(CEMENT, TARGET)$ values from *Concrete* dataset (see Section 3) have been used to this end:

- The graphs in the first row show, respectively, the pairs of points associated with the original variables $(X_j, Y)$, those transformed through the corresponding marginal cumulative distribution function $((U_j, V) = F_{X_j}(X_j), G_Y(Y))$, and the density $c^*$ associated with the copula $C^*$ that best fits the latter.
- On the other hand, in the first graph of the second row, the value $x_j$ of the variable *CEMENT*, to which the value of the variable *TARGET* is conditioned, is marked with a red vertical reference. Next, the value $u_j$ transformed by the cumulative distribution function $F_{X_j}$ is marked as well. Finally, the last graph shows the conditional density copula $c^*(v|u_j)$ associated with this value. This function is used to weight the values of variable *TARGET* to estimate $\hat{Y}_{(k=1)} = E[Y|X_j^* = x_j]$.

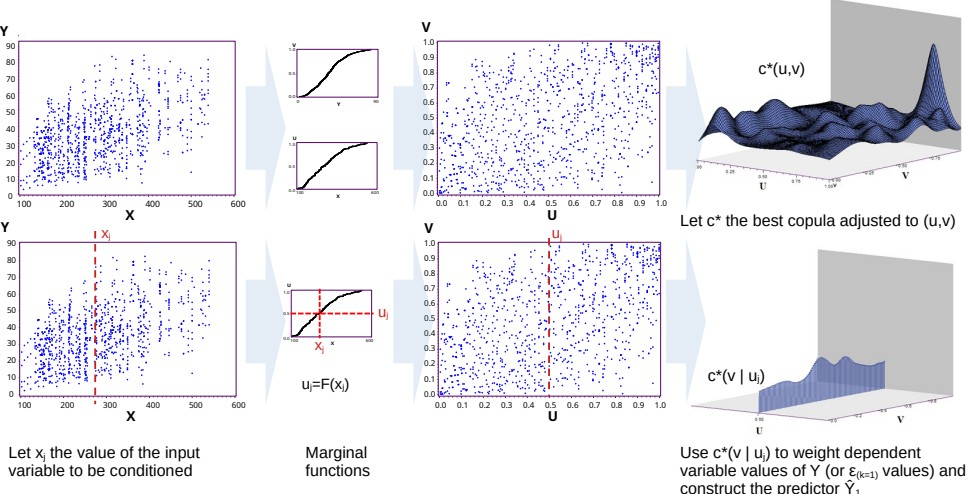

**Figure 2.** Construction of an ADABOC predictor.

The following step ($k = 2$) starts again with the search of the pair $(C^*_{j,2}, X^*_j)$ that provides a lower AIC in the adjustment process to the values $(u_j(i), v_{k=2}(i))$. These ones are obtained from $(x^*_j(i), \hat{\varepsilon}_{k=2}(i))$, through their respective cumulative distribution functions.

Then, the expected value of $\varepsilon_{(k=2)}$ must be estimated for each value of $X^*_j$:

$$\hat{\varepsilon}_{(k=2)} = E[\varepsilon_{(k=2)}|X^*_j = x^*_j] \tag{17}$$

A new predictor for the dependent variable $Y$ will be generated at the end of this step:

$$\hat{Y}_{(k=2)} = \hat{Y}_{(k=1)} + \hat{\varepsilon}_{(k=2)} = \hat{Y}_{(k=0)} + \hat{\varepsilon}_{(k=1)} + \hat{\varepsilon}_{(k=2)} = \bar{Y} + \hat{\varepsilon}_{(k=1)} + \hat{\varepsilon}_{(k=2)} \tag{18}$$

The algorithm stops when the number of iterations specified by the *maxiter* parameter is reached, unless the early stopping criterion is satisfied or the product copula $\pi$ is selected as the best one in some of the iterations.

Observe the fact that the same input variable can be selected repeatedly to explain the resultant error terms from different iterations. Note that some algorithms, such as random forest [43] or gradient boosting [39], allow this kind of circumstance. In fact, they allow the same variable not only to participate in different trees, but to be used several times in the same tree.

Note too that the predictors adjustment process is carried out sequentially, and so each one of them is conditioned by the order of selection of the explanatory variables—that is, it will not obtain the same result by predicting $Y$ as a function of $X_1$ and the resulting residual from $X_2$, by predicting $Y$ as a function of $X_2$, and the resulting residual from $X_1$. In fact, neither the family of copulas selected for each one of these variables proceeding in one way or another nor the estimation of the parameter they depend on would be the same.

Alternatively, the predictor could be built by using, simultaneously, all the explanatory variables at a time, through a $(m + 1) - copula$ that would link the target variable $Y$ with the $m$ independent ones $X_j$. This way, the same input variable could not enter more than once in the model and the multidimensional parameter estimation the copula depends on would be carried out jointly, as occurs, for example, in a regression model. However, as it has been pointed in Section 1.1 that attempting to capture dependency structures using a single multivariate parametric copula could be an arduous task. The fact that the variety of the studied m-dimensional copula functions, when $m$ is greater than 2, is significantly smaller than those available in the bidimensional case, which is the most referenced in the literature, must be taken into account. This would reduce the spectrum of

them available to identify the dependency relationship and, as a consequence, the accuracy of the predictions.

Once the training process has concluded, the predictors generated in the different iterations are compared with respect to the MAE metric. This one is measured over the validation dataset to choose the best estimator.

Thus, the final predictor will present the expression:

$$\hat{Y}_{k*} = \bar{Y} + \sum_{k=1}^{k*} \hat{\varepsilon}_k \tag{19}$$

where $k*$ is the iteration in which the best value of the MAE metric over the validation dataset has been achieved.

The described process has been coded in Algorithm 1. The output returned by this algorithm (see Equation (20)) is an object that contains the predictor $\hat{Y}_{k*}$ and the information needed to score any dataset that contains the same input variables as those used in the adjustment process. Algorithm 2 details this scoring process.

$$\{(X_{j_1}, F_{X_{j_1}}, C_1^*, data_1), (X_{j_2}, F_{X_{j_2}}, C_2^*, data_2), ..., (X_{j_{k*}}, F_{X_{j_{k*}}}, C_{k*}^*, data_{k*})\} \tag{20}$$

The object defined by Equation (20) consists of a list of 4-tuples, where the components of the $k$th one are:

1.  $X_{j_k}$—the $k$th input variable entered in the $k$th step to explain $\varepsilon_k$. Note that subscripts $j_k$ could refer to the same variable for two distinct iterations, meaning that the same variable can participate in different steps;
2.  $F_{X_{j_k}}$—the cumulative distribution function associated with $X_{j_k}$. This function is used to transform an $x_{j_k}$ value into a $u_{j_k}$ value;
3.  $C_k^*$—the copula function adjusted to $(u_{j_k}, v_k)$ in the $k^{th}$ iteration;
4.  $data_k = (x_{j_k}(i), \varepsilon_k(i))$—the data associated with all the observations "$i$" of training and validation datasets. This information is useful to avoid repeating the calculus of $\varepsilon_k$ with copula function $C_k^*$ (through the corresponding conditional function) for values of $X_{j_k}$ used in the adjustment process and for values of $X_{j_k}$ contained in training and validation datasets. So, the second component $F_{X_{j_k}}$, will be used only in the case where during the scoring process a value of $X_{j_k}$ different to the values registered in these samples appears. This allows one to reduce the computational time cost involved in the scoring process of a new dataset.

So, the predictor is by constructed decomposing the dependent variable in a relationship of error terms that have been estimated from copula functions. This is the reason the proposed method has been named the *Additive Decomposition Algorithm Based On Copulas* (or ADABOC).

$$\hat{Y}_4 = \bar{Y} + \hat{\varepsilon}_1 + \hat{\varepsilon}_2 + \hat{\varepsilon}_3 + \hat{\varepsilon}_4 \tag{21}$$

By way of illustration, Figure 3 shows the MAE variation carried out by Algorithm 1 along the validation sample associated with the *Concrete* dataset (see Section 3). The pair copula $(variable, copula) = (X_{j_k}, C_k^*)$ that has been selected in each step k is specified in the boxes. So, for example, the variable named *CEMENT* and a *SurvivalBB*1 copula have been selected in the first iteration, the variable named *SUPERPLASTICIZER* and a *Frank* copula have been selected in the second one and so on.

---

**Algorithm 1:** ADABOC.

---

**Inputs:** $Y$ Target variable

$X_1, X_2, ..., X_m$ Input variables

$\{y(i), x_1(i), x_2(i), ..., x_m(i)\}$ Data observations, $i = 1, 2, ..., n$

$n_1$ Number of training observations

$n_2$ Number of validation observations

*subsamplePercent* Percent of the $n_1$ train observations randomly selected for iteration

*maxIter* Maximum number of iterations

*earlyStoppingIterations* Maximum number of iterations without an improvement

*epsilon* Precision error

$\{C^r\}$ Families of copulas, $r = 1, 2, ..., p$

$n = n_1 + n_2$

$subn_1 =$ round($subsamplePercent * n_1$) Num of train observts. randomly selected for iteration

*counterNoImprovement* Cumulative number of iterations without improvement counter

$\hat{y}_0(i) = \frac{1}{n_1} \sum_{s=1}^{n_1} y_s, \forall i = 1, 2, ..., n$

$\hat{\varepsilon}_{(1)}(i) = y(i) - \hat{y}_0(i), \ \forall i = 1, 2, ..., n$

$MAE_1 = \frac{1}{n_d} \sum_{i=1}^{n_d} |\hat{\varepsilon}_{(1)}(i)|$ On validation dataset if it exists (d=2); else on training dataset (d = 1)

$k = 0$ Iterations counter

**while** *(k ≤ maxIter) and (counterNoImprovement ≤ earlyStoppingIterations)* **do**

> $k = k + 1$
>
> $F_{X_j} = ker\_cdf(x_j) \forall j = 1, 2, ..., m$ Estimate $F_{X_j}$ distrib. from train data by kernel estimators
>
> $G_{\varepsilon_k} = ker\_cdf(\varepsilon_k)$ Estimate $G_{\varepsilon_k}$ distribution from training data using kernel estimators
>
> $(u_j(i), v_k(i)) = \{(F_{X_j}(x_j(i)), G_{\varepsilon_k}(\hat{\varepsilon}_k(i)))\} \forall i = 1, 2..., n \ \forall j = 1, 2..., m$ Pair in copula dom.
>
> $(C_k^*, X_{j_k}) = \text{argmin}_{j,r}\{AIC(C_{j,k}^r)\}$ The minimum and the variable in which it is reached
>
> $AIC(C_{j,k}^r) = \text{fitVarCop}(C^r, subn_1, (u_{j_k}, v_k))$ AIC by fit copula to $subn_1$ rand pairs of train
>
> **if** $C_k^* = \Pi$ Independence criterion
>
> > **then**
> >
> > > $k = maxIter + 1$
> >
> > **else**
> >
> > > $E_k =$
> > >
> > > **for** *i=1 to n* **do**
> > >
> > > > $E_\varepsilon = E[\varepsilon_k | X_{j_k}^* = x_{j_k}(i)]$ Use $g_{\hat{\varepsilon}_{(k=1)}}^{-1}$ & $c_{k|1}^*$, density functs. with $G_{\varepsilon_k}$ & copula $C_k^*$
> > > >
> > > > $E_k[X_{j_k}, x_{j_k}(i)] = E_\varepsilon$ For each input we store needed data for scoring other sets
> > > >
> > > > $\hat{y}_k(i) = \hat{y}_{k-1}(i) + E_\varepsilon$ Calculate the new Y estimator
> > > >
> > > > $\hat{\varepsilon}_{k+1}(i) = y(i) - \hat{y}_k(i)$ Calculate the new error variable
> > >
> > > $MAE_{k+1} = \frac{1}{n_2} \sum_{l=1}^{n_2} |\hat{\varepsilon}_{k+1}(l)|$ On validation set if it exists (else over training set)
> > >
> > > **if** *round($MAE_{k+1}$, epsilon) < round($MAE_k$, epsilon)* Early stopping criterion
> > >
> > > > **then**
> > > >
> > > > > *counterNoImprovement* $= 0$
> > > > >
> > > > > $k^* = k$ Index associated with the iteration with minimum MAE
> > > >
> > > > **else**
> > > >
> > > > > *counterNoImprovement* $=$ *counterNoImprovement* $+ 1$

*copulaModel* $=$

$\{\hat{Y}_0 = \bar{Y}_{n_1}, (X_{j_1}, F_{X_{j_1}}, C_1^*, data_1), (X_{j_2}, F_{X_{j_2}}, C_2^*, data_2), ..., (X_{j_{k^*}}, F_{X_{j_{k^*}}}, C_{k^*}^*, data_{k^*})\}$

**return** *copulaModel*

---

The algorithm stopped in the 28th iteration as the best copula selected in this step has been the product copula $\pi$. It means that the residual process resulting after 28 iterations is independent with respect to all the explanatory variables. So, as the minimum MAE value calculated over the validation table has been achieved in iteration 21 (red point in Figure 3), the final predictor is composed as follows:

$$\hat{Y}_{k^*} = \hat{Y}_{21} = \bar{Y} + \sum_{k=1}^{21} \hat{\varepsilon}_k \qquad (22)$$

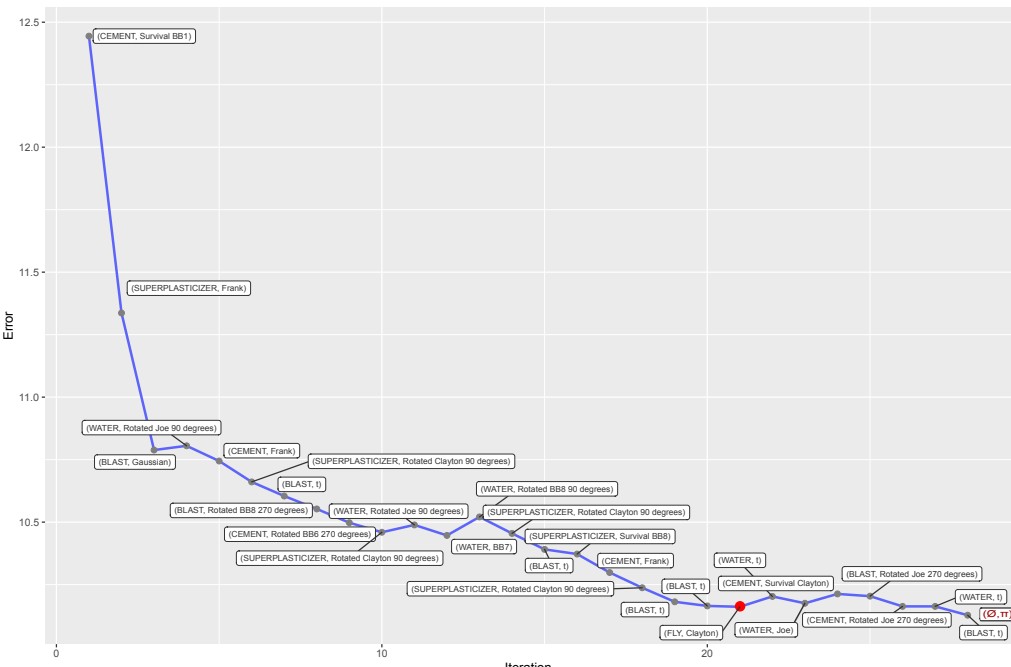

**Figure 3.** Evolution of MAE metric using ADABOC over a validation dataset.

*2.3. Implementation Details*

ADABOC have been implemented in *R* software through two main functions (see github.com/jfvelezserrano/ADABOC, accessed on 25 May 2021):

1. Function *ADABOC.R* implements Algorithm 1. It depends on the next input parameters:

   - trainDataset (required): name of the dataframe associated with the training dataset.
   - validationDataset (optional): name of the dataframe associated with the validation dataset. If this value is omitted, then the predictor returned by the algorithm is generated in the iteration specified by the *maxiter* parameter;
   - testDataset (optional): name of the dataframe associated with the test dataset. In case a dataset is provided, the value of the metric specified by the evalMetric parameter is calculated over it;
   - evalMetric (required): metric to be evaluated. It can take the values: MAE, MAPE, MedAPE, SMAPE, MSE, RMSE. By default: MAE;
   - maxiter (required): maximum number of iterations carried out by the algorithm. By default: 200;
   - subsamplePercent (optional): observed percentage of the training dataset randomly selected in each iteration for the adjustment process. If this value is omitted, then all the observations of the training dataset are used;
   - earlyStoppingIterations (optional): maximum number of iterations allowed without an improvement of the metric specified by the evalMetric parameter. If this value is omitted, then the algorithm does not stop early stopping criterion;
   - epsilon (optional): a precision parameter that identifies the number of decimal positions considered in the measurement of the evalMetric parameter. It allows one to evaluate if this metric improves from one step to the next one. By default: 14;
   - numBins (optional): number of equidistant points generated for approximating the calculus involved in the estimation of the density function $c_{k|1}^*$. By default: 2000;
   - modelCopula: output returned by the model. It is an object of the type referenced in Equation (20).

2. Function *ADABOCPredict.R* implements Algorithm 2. It depends on the next input parameters:

   - model (required): name of the object returned by the *ADABOC.R* function;
   - scoreDataset (required): name of the dataframe associated with the dataset to be scored. It must contain the input variables involved in the predictor given by the previous parameter;
   - prediction: output returned by the function. It contains the predictions obtained for each one of the observations *i* stored in scoreDataset. Observe that the expected values are given by $\hat{\varepsilon}_{(k)}(i) = E[\varepsilon_{(k)}|X_j^* = x_j^*(i)]$, which must only be calculated for the values $x_j^*(i)$ not registered in *trainingDataset* or *tvalidationDataset*. So, the estimated ones already generated by the *ADABOC* function during the adjustment process are used.

---

**Algorithm 2:** Score function associated with ADABOC.

**Inputs:** $\{\hat{Y}_0 = \bar{Y}_{n_1}, (X_{j_1}, F_{X_{j_1}}, C_1^*, data_1), ..., (X_{j_{k^*}}, F_{X_{j_{k^*}}}, C_{k^*}^*, data_{k^*})\}$ From ADABOC

$scoreDataset = \{X_{j_k}^{score}\}$ Dataset to be scored ($X_{j_k}$ variables must be in copulaModel)

**for** *i = 1 to rows(scoreDataset)* **do**

   $\hat{y}_0(i) = \hat{Y}_0$

   **for** *k=1 to k** **do**

      **if** $E_k.contains(X_j, x_{j_k}^{score}(i))$ **then**

         $errorEstimator = E_k[X_j, x_{j_k}^{score}(i)]$

      **else**

         $u_{j_k}^{score}(i) = F_{X_{j_k}}(x_{j_k}^{score}(i))$

         $errorEstimator = E[\varepsilon_k|X_{j_k}^* = x_{j_k}^{score}(i)]$ Use $g_{\varepsilon_{(k=1)}}^{-1}$ and $c_{k|1}^*$

      $\hat{y}_k(i) = \hat{y}_{k-1}(i) + errorEstimator$

**return** $\hat{y} = [\hat{y}_{k^*}(1), \hat{y}_{k^*}(2), ..., \hat{y}_{k^*}(n_s)]$

---

## 3. Results

In this chapter, ADABOC and other competitive machine learning models are compared over some typical datasets.

### 3.1. Datasets Used

The datasets used have been extracted from Knowledge *UCI: Machine Learning Repository* (archive.ics.uci.edu/ml/datasets, accessed on 25 May 2021), imposing some filters on the search criteria:

- The dependent variable must be of the interval type but there must not be a temporary dependence between its values. ADABOC is not oriented to the modeling of time series.
- Datasets must be big enough to make it reasonable to apply machine learning methods. Thus, the number of rows of the dataset must be between 1000 and 10,000 and the number of attributes must be higher than 5.

Taking into account these filters, the datasets considered are shown in Table 2. Next, some observations are made:

- Dataset 2 has two target variables: next-day maximum and minimum air temperatures. To consider only one problem associated with this dataset, next-day medium air temperature has been considered. This one has been calculated as the average of the two target variables.
- Dataset 8 has two target variables too: motor_UPDRS and total_UPDRS. The relative positions between the different tested models are similar with one another, which is the reason why only one of them (total_UPDRS) has been considered.
- Dataset 4 is the same that Dataset 3 but adding two additional input variables and not normalizing data. That is the reason why only the first one has been considered.

- Dataset 7, available in the *UCI repository* associated with the problem *KDD Cup 1998 Data*, has been replaced by the one available in www4.stat.ncsu.edu/~dickey/Analytics/ Datamine/data (accessed on 25 May 2021) as the number of input variables of this dataset is considerably lower, allowing the reduction in the computational cost of the proof tested while keeping a good predictive capacity (with this dataset, the SAS Institute achieved the second position in the KDD 1998 competition).
- Each one of the datasets has been split into training, validation and test samples, with 40%, 30% and 30% of the observations for each one, respectively. The corresponding sizes are shown.

**Table 2.** Datasets selected from UCI repository.

| Dataset | Inputs | Total (Rows) | Training (Rows) | Validation (Rows) | Test (Rows) |
|---|---|---|---|---|---|
| Airfoil (1) | 6 | 1503 | 601 | 451 | 451 |
| Bias (2) | 21 | 7590 | 3036 | 2277 | 2277 |
| Communities (3) | 100 | 1994 | 798 | 598 | 598 |
| CommunitiesU (4) | 102 | 1994 | 798 | 598 | 598 |
| Concrete (5) | 8 | 1030 | 412 | 309 | 309 |
| Electrical (6) | 12 | 10,000 | 4000 | 3000 | 3000 |
| KDD1998 (7) | 18 | 4843 | 1937 | 1453 | 1453 |
| Parkinsons (8) | 16 | 5875 | 2351 | 1762 | 1762 |

In addition to the filters imposed, some of the variables of the datasets have been dropped:

- On the one hand, the ones with missing values were removed. This is justified by the nature of some of the models used in the comparison. For example, random forest and gradient boosting models allow these kinds of values. However, ADABOC as well as some of the other models taken into account in the comparative, such as GLM and neural networks, need the imputation of the mentioned values, with a lot of methods available for this task: to impute by using the mean or the median value, to impute according to the distribution of the variable, predict them, etc. Thus, with the aim of isolating the effect that these methods could have on the final results obtained, these variables have been removed.
- On the other hand, the ones of nominal nature (factors) were removed. This is justified by the convenience of not making inferences with copulas using non-continuous variables (see Section 2.1).

### 3.2. Experiments

Algorithm 1 has been applied to the datasets described in the previous section. It could use any type of copula functions. Specifically, taking into account that the algorithm has been implemented depending on *BiCopEst* (Parameter Estimation for Bivariate Copula Data) R-function of the *VineCopula* package to select the copula with best AIC value, the ones contained in this function have used (see Table A1 in Appendix A). Only Tawn's copulas have been excluded as they severely increased the computational time cost without any noticeable improvement in the results achieved.

Regarding the parameters of the function itself, all of them take the default values presented in Section 2.2. Due to the fact that there is a validation table, the *earlySopping Iterations* parameter has been used. Its value has been set to 10 iterations. Additionally, several values of the *subsamplesize* parameter have been tested, from 10 to 100 with a *bystep* value equal to 10. Table 3 allows comparing of MAE results over validation table, according to the variation of this parameter. In general, it can be noted that there are no significant differences between them and that higher sizes do not provide better results. This is an important observation as it allows reducing of the computational time cost of the model, making use of small samples to train it.

**Table 3.** ADABOC MAE results by subsample parameter value.

| Subsample | Airfoil | Bias | CommunitiesU | Concrete | Electrical | KDD1998 | Parkinsons |
|---|---|---|---|---|---|---|---|
| 10% | 3.829 | 0.8007 | **235.3** | 11.04 | 0.0145 | 4.149 | 8.274 |
| 20% | 3.637 | 0.8102 | 235.9 | 10.52 | 0.0141 | 4.159 | 8.236 |
| 30% | 3.730 | **0.7805** | 250.9 | 10.54 | 0.0142 | 4.115 | 8.106 |
| 40% | 3.632 | 0.7847 | 245.5 | 10.34 | 0.0142 | 4.133 | 8.057 |
| 50% | 3.614 | 0.8437 | 263.2 | 10.26 | **0.0139** | 4.152 | **7.964** |
| 60% | 3.604 | 0.8457 | 253.1 | 10.43 | 0.0146 | 4.143 | 8.080 |
| 70% | 3.732 | 0.8503 | 242.6 | 10.26 | 0.0141 | 4.101 | 8.099 |
| 80% | 3.579 | 0.8660 | 240.4 | **10.07** | 0.0141 | 4.145 | 8.183 |
| 90% | 3.579 | 0.8674 | 249.8 | 10.48 | 0.0146 | **4.094** | 8.202 |
| 100% | **3.5514** | 0.8622 | 239.4 | 10.42 | 0.0144 | 4.104 | 8.242 |

Then, the predictor associated with the best results obtained for this parameter over each validation dataset (highlighted in bold in Table 3) was applied over the test sample. The corresponding results are the ones presented in the row identified as ADABOC in Table 4. It can be observed that theses values are similar to those obtained over the validation table, avoiding overfitting effects.

**Table 4.** MAE results by algorithm.

| Method | Airfoil | Bias | CommunitiesU | Concrete | Electrical | KDD1998 | Parkinsons |
|---|---|---|---|---|---|---|---|
| Intercept | 5.148 | 2.0481 | 482.4 | 13.51 | 0.0309 | 7.443 | 9.091 |
| GBM | 1.313 | 0.5386 | 273.2 | 10.10 | 0.0066 | 4.083 | 6.827 |
| GLM | 3.569 | 0.7860 | 310.0 | 10.42 | 0.0174 | 4.193 | 8.380 |
| ADABOC | 3.339 | 0.7696 | **270.1** | 10.16 | 0.0138 | **3.837** | 8.198 |
| NN | 1.729 | **0.4901** | 287.3 | **9.32** | **0.0048** | 4.071 | **6.369** |
| DRF | 2.800 | 0.5922 | 277.9 | 10.16 | 0.0093 | 4.109 | 6.915 |
| XGB | **1.283** | 0.6202 | 287.1 | 10.90 | 0.0077 | 4.516 | 7.298 |

On the other hand, some popular machine learning models have been adjusted to compare the results they provide with ADABOC. To this end, *h2o.autoML* function, which invokes to $H_2O$ platform, has been used. This function adjusts the following type of models:

- GBM—Gradient Boosting Model (A);
- GLM—General Linear Model (B);
- NN—Neural Network, identified as *Deep Learning* models in $H_2O$ (C);
- DRF—Distributed Random Forest (D);
- XGB—Extreme Gradient Boosting (E).

Tuning of the parameters associated with each of these models was carried out by *h2o.autoML* function. For instance, in GBM, tree depth and shrinkage parameters are allowed to vary on a grid of values, while in NN different architectures are tested.

Additionally, the *h2o.autoML* function allows specification of the metric to be optimized along the adjustment process of each model over the validation table (specified through the *validation_frame* parameter). That metric is specified through the *stopping_metric* parameter (MAE has been used). Results, obtained with the best model identified for each one of the mentioned types, are compared over the test dataset (specified through the *leaderboard_frame* parameter) in Table 4. The results corresponding to the winner model are remarked in bold.

It should be noted that these results were obtained with the execution of the *h2o.autoML* function conditioned by the value of a random seed whose value was set to 12345. Fixing the value of this seed allows the results to be replicated. However, the documentation of this function aims for replication to not be possible if NNs are included in the modeling process. For this reason, two executions of this function have been made:

- The first of them excludes NN models. In this one, it has been specified that the function adjusts a maximum of 10 models. According to these considerations, in each execution, 5 GBM, 3 XGB, and 1 of each of the other types mentioned (GLM and DRF) are compared (see Table 5).
- The second one includes only NN models. The number of them have been fixed to 10 (see Table 6).

A reasonably high value has been specified in the computation time of each of these executions, which have guaranteed the complete execution thereof. This value have been specified through the *max_runtime_secs* parameter of the *h2o.autoML* function. It was set to 10,800 s (3 h) in the first execution and to 25,200 (7 h, although they were not spent) in the second. The fact that it has been significantly greater in the latter is justified by the high computational time needed by the neural network models.

**Table 5.** Top 10 machine learning models excluding neural network models.

| Rank | Airfoil | Bias | CommunitiesU | Concrete | Electrical | KDD1998 | Parkinsons |
|---|---|---|---|---|---|---|---|
| 1 | 1.283 (E) | 0.5386 (A) | 273.2 (A) | 10.10 (A) | 0.0066 (A) | 4.083 (A) | 6.827 (A) |
| 2 | 1.313 (A) | 0.5424 (A) | 274.4 (A) | 10.13 (A) | 0.0070 (A) | 4.109 (D) | 6.968 (A) |
| 3 | 1.314 (A) | 0.5477 (A) | 277.9 (A) | 10.14 (A) | 0.0070 (A) | 4.155 (A) | 7.016 (A) |
| 4 | 1.349 (A) | 0.5558 (A) | 277.9 (D) | 10.16 (D) | 0.0072 (A) | 4.181 (A) | 7.022 (D) |
| 5 | 1.362 (A) | 0.5763 (A) | 279.6 (A) | 10.34 (A) | 0.0077 (A) | 4.193 (B) | 7.022 (A) |
| 6 | 1.379 (E) | 0.5922 (D) | 285.5 (A) | 10.42 (B) | 0.0077 (E) | 4.211 (A) | 7.233 (A) |
| 7 | 1.397 (E) | 0.6202 (E) | 287.1 (E) | 10.54 (A) | 0.0081 (E) | 4.270 (A) | 7.428 (E) |
| 8 | 2.758 (A) | 0.6288 (E) | 294.7 (E) | 10.90 (E) | 0.0087 (E) | 4.516 (E) | 7.495 (E) |
| 9 | 2.800 (D) | 0.6298 (E) | 299.7 (E) | 11.32 (E) | 0.0093 (D) | 5.014 (E) | 7.730 (E) |
| 10 | 3.569 (B) | 0.7860 (B) | 310.0 (B) | 11.37 (E) | 0.0174 (B) | 5.024 (E) | 8.380 (B) |

**Table 6.** Top 10 neural network models.

| Ranking | Airfoil | Bias | CommunitiesU | Concrete | Electrical | KDD1998 | Parkinsons |
|---|---|---|---|---|---|---|---|
| 1 | 1.729 | 0.4901 | 287.3 | 9.326 | 0.0048 | 4.071 | 6.369 |
| 2 | 1.763 | 0.4967 | 294.4 | 9.358 | 0.0053 | 4.137 | 6.471 |
| 3 | 1.834 | 0.5005 | 294.4 | 9.620 | 0.0066 | 4.146 | 6.690 |
| 4 | 1.956 | 0.5721 | 297.8 | 9.954 | 0.0095 | 4.277 | 6.915 |
| 5 | 2.124 | 0.5889 | 299.1 | 10.115 | 0.0101 | 4.277 | 7.114 |
| 6 | 2.356 | 0.6174 | 307.3 | 10.195 | 0.0102 | 4.342 | 7.273 |
| 7 | 2.759 | 0.6324 | 308.5 | 10.616 | 0.0106 | 4.523 | 7.298 |
| 8 | 2.887 | 0.7517 | 321.3 | 10.640 | 0.0106 | 4.630 | 7.381 |
| 9 | 3.305 | 0.9060 | 324.6 | 10.645 | 0.0127 | 4.762 | 7.510 |
| 10 | 3.418 | 1.1164 | 335.2 | 10.692 | 0.0149 | 4.910 | 7.841 |

In Table 4, results corresponding to the best modeling strategy have been highlighted in bold. A baseline associated with the MAE of predictions is given by the intercept term of a regression model only, and the average of the target variable over the training dataset is shown too.

It is important to remark that the aim is to compare ADABOC with other well known machine learning methods. A previous phase of processing data would allow better results to be achieved with any of them. However, this phase was been carried out as the purpose is to test if the algorithm is competitive versus other techniques, when all of them are trained and tested with common datasets.

## 4. Discussion

According to the results shown in Table 4, ADABOC seems to be a competitive machine learning algorithm. In fact:

- It provides the best results in 2 of the 7 forecasting problems, *CommunitiesU* and *KDD*1998, even improving the ones obtained with NN and GBM, the best models globally.
- In the dataset *Concrete*, it improves the results given by DRF, XGB and GLM models, , which are not far from the other ones. In fact, it is superior to two of the GBM models and five of the adjusted NN models (see Tables 5 and 6, respectively).
- It is always better than GLM, which is the worst model, although the predictions of the latest cause the MAE given by only an intercept term to reduce. Thus, in any case, it can be considered as a better alternative to a linear model.

Not in vain, the ADABOC methodology incorporates some of the modeling strategies used by other well known machine learning techniques such as the use of an early stopping criterion or the use of subsamples of the training dataset in each one of the steps of the iterative process. Next, other relevant points in common with some of these methods are presented:

- On the one hand, it can be observed that the same input variable can be selected repeatedly to explain the resultant error terms from different iterations. Note that some of the algorithms, such as random forest [43] or gradient boosting [39], allow this kind of circumstances. In fact, they allow the same variable to not only to participate in different trees but to do it several times in the same tree. In addition, in the proposed algorithm, it could be allowed that, in each one of the iterations, $p$ ($\leq m$) explanatory variables were randomly selected as models, as in random forests. This way, it would avoid those with greater predictive power being repeatedly selected, in favor the rest of them.
- On the other hand, it can be observed that the predictor adjustment process was carried out sequentially, and so each one of them was conditioned by the order of selection of the explanatory variables—that is, the same result will not be obtained by predicting $Y$ as a function of $X_1$ and the resulting residual from $X_2$, by predicting $Y$ as a function of $X_2$ and the resulting residual from $X_1$. In fact, neither the family of copulas selected for each one of these variables proceeding in one way or another nort the estimation of the parameter they depend on would be the same.
  Alternatively, the predictor could be built by using, simultaneously, all the explanatory variables at a time, through a $(m+1) - copula$ that would link the target variable $Y$ with the $m$ independent ones $X_j$. This way, the same input variable could not be entered more than once in the model and the multidimensional parameter estimation the copula depends on would be carried out jointly, as occurs, for example, in a regression model. However, as it has been pointed in Section 1.1, attempting to capture dependency structures using a single multivariate parametric copula could be an arduous task.
  As an intermediate possibility, making an adjustment based on 3-copula functions could be considered,. This means that, in each iteration, the copula would relate each error term, $\varepsilon_k$, with a couple of explanatory variables. This way, possible effects associated with interactions between variables could be reflected in the same way that would be achieved with the regression techniques or with the ensemble modeling mentioned when handling decision trees with depth values higher than 1.
  It must be taken into account that the variety of m-dimensional copula functions studied when $m$ is greater than 2 is significantly smaller than those available in the bidimensional case, which is the most referenced in the literature. This would significantly reduce the spectrum of those available to identify the dependency relationship and, in consequence, the accuracy of the predictions.

Finally, it must be taken into account that the predictor generated by the Algorithm 1 could be combined through a *stacking* strategy with the ones provided by other techniques, to try to achieve an improvement of the results.

## 5. Conclusions

In this paper a novel methodology for forecasting interval variables based on the use of bivariate copula functions is proposed. The core of the methodology is an iterative algorithm that identifies those bicopulas that best fit the dependency relationships between the independent variables and the different error terms obtained during the adjustment process. The method has been tested on several known datasets, establishing a comparison with other popular and competitive machine learning techniques, achieving satisfying results.

Nowadays, the authors are working on several lines of improvement of this algorithm:

- First of all, ensuring the ability to handle variables of a discrete nature, in a similar way that the vine copula regression method proposed in [29] do;
- Secondly, the possibility of assigning higher weights to higher value errors, in a similar way that boosting methods do  to improve the training of the method.
- Thirdly, the capacity to include analysis of the interactions between the independent variables use in the model. The proposed method has already shown the concept of interaction since the predictor mixes different explanatory variables in its definition. However, the use of 3-copula functions that would allow relating pairs of independent variables with the successive error terms obtained during the adjustment could be an interesting possibility [44] as it would capture more complex dependency relationships and possibly improve the predictive capacity of the method.
- Finally, although the number of problems in which there is not a single dependent variable is significantly lower, the possibility of adapting it to problems in which there is a vector of dependent variables is also being evaluated. In this regard, the concept of conditional copula associated with the variable $(X, Y) \mid W$ introduced by [45], which would allow the prediction of the pair $(X, Y)$ based on the value of a dependent variable $W$, is being considered.

The final aim is that ADABOC achieves more competitive results and works regardless of the amount and the nature of exogenous and endogenous variables involved in the model.

**Author Contributions:** Conceptualization, D.V.; methodology, D.V. and J.A.C.; software, J.A.C., J.F.V., M.N. and D.V.; validation, J.A.C., M.N. and D.V.; formal analysis, D.V. and J.A.C.; investigation, D.V.; resources, J.F.V., D.V., J.A.C. and M.N.; data curation, M.N., J.A.C. and D.V.; writing—original draft preparation, J.F.V. and D.V.; writing—review and editing, J.F.V., D.V. and M.N.; visualization, J.F.V. and D.V.; supervision, J.F.V., D.V. and M.N.; project administration, J.F.V.; funding acquisition, J.F.V. and D.V. All authors have read and agreed to the published version of the manuscript.

**Funding:** This research was funded by the Spanish Ministry of Science and Innovation, under the "RETOS" Program, grant number: RTI2018-098019-B-I00; Spanish Ministry of Science, Innovation and Universities, under the "I + D + i" Program, grant number: PID2019-106254RB-I00; and by the CYTED Network "Ibero-American Thematic Network on ICT Applications for Smart Cities", grant number: 518RT0559.

**Data Availability Statement:** Public datasets used in experimental section: archive.ics.uci.edu/ml/datasets (accessed on 25 May 2021); Reproducible test code of ADABOC: github.com/jfvelezserrano/ADABOC (accessed on 25 May 2021).

**Conflicts of Interest:** The authors declare no conflict of interest. The funders had no role in the design of the study; in the collection, analyses, or interpretation of data; in the writing the manuscript, or in the decision to publish the results.

## Abbreviations

The following abbreviations are used in this manuscript:

| | |
|---|---|
| ADABOC | Additive Decomposition Algorithm Based On Copulas |
| AIC | Akaike Information Criterion |
| BIC | Bayesian Inference Criterion |
| DRF | Distributed Random Forest |
| GBM | Gradient Boosting Model |
| GLM | General Linear Model |
| MAE | Mean Absolute Error |
| MAE | Mean Absolute Percentage Error |
| MedAPE | Median Absolute Percentage Error |
| MSE | Mean Squared Error |
| NN | Neural Network |
| RMSE | Root Mean Square Error |
| SMAPE | Symmetric Mean Absolute Percentage Error |
| XGB | Extreme Gradient Boosting |

## Appendix A. Copula Families Used by ADABOC

**Table A1.** Copula families of *VineCopula* R-package considered in ADABOC.

| Id Family | Name |
|:---:|:---:|
| 0 | Independence copula ($\pi$) |
| 1 | Gaussian copula |
| 2 | Student t copula (t-copula) |
| 3 | Clayton copula |
| 4 | Gumbel copula |
| 5 | Frank copula |
| 6 | Joe copula |
| 7 | BB1 copula |
| 8 | BB6 copula |
| 9 | BB7 copula |
| 10 | BB8 copula |
| 13 | rotated Clayton copula (180 degrees; "survival Clayton") |
| 14 | rotated Gumbel copula (180 degrees; "survival Gumbel") |
| 16 | rotated Joe copula (180 degrees; "survival Joe") |
| 17 | rotated BB1 copula (180 degrees; "survival BB1") |
| 18 | rotated BB6 copula (180 degrees; "survival BB6") |
| 19 | rotated BB7 copula (180 degrees; "survival BB7") |
| 20 | rotated BB8 copula (180 degrees; "survival BB8") |
| 23 | rotated Clayton copula (90 degrees) |
| 24 | rotated Gumbel copula (90 degrees) |
| 26 | rotated Joe copula (90 degrees) |
| 27 | rotated BB1 copula (90 degrees) |
| 28 | rotated BB6 copula (90 degrees) |
| 29 | rotated BB7 copula (90 degrees) |
| 30 | rotated BB8 copula (90 degrees) |
| 33 | rotated Clayton copula (270 degrees) |
| 34 | rotated Gumbel copula (270 degrees) |
| 36 | rotated Joe copula (270 degrees) |
| 37 | rotated BB1 copula (270 degrees) |
| 38 | rotated BB6 copula (270 degrees) |
| 39 | rotated BB7 copula (270 degrees) |
| 40 | rotated BB8 copula (270 degrees) |

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
