# Peer review of "A New Machine Learning Forecasting Algorithm Based on Bivariate Copula Functions"

_forecasting, doi:10.3390/forecast3020023_

Round 1

Reviewer 1 Report

This is an interesting topic. I have several questions on this paper which need clarification prior to getting the "green light" for acceptance. Please note and address them below.

(1) The risk references are incomplete. Please add some more modern references to make the paper more accurate such as the following:

Guharay, Sabyasachi, K. C. Chang, and Jie Xu. "Robust Estimation of Value-at-Risk through Distribution-Free and Parametric Approaches Using the Joint Severity and Frequency Model: Applications in Financial, Actuarial, and Natural Calamities Domains." Risks 5, no. 3 (2017): 41.

(2) Line 188: "Selection of the copula function C which best models the
relationship between explanatory variables and the residual term obtained in the previous step." What are the range of copula functions which are allowed? There are many types including empirical copulas. So what are the eligible ones?

(3) From Figure 1: Why is the Expected value used solely? For data with high outliers, why isn't Median being used?

(4) Equation (12) notation for d_epsilon is not clear.

(5) For Table 1, why is AIC being chosen instead of BIC?

(6) Figure 3 is based on exactly what? A simulation example? Real life data? Further clarity is needed.

(7) In Algorithm 2, what is the precise definition of the auxError?

(8) For the Results section: Are there are any simulated datasets where the results are known? This is crucial in the Verification & Validation steps. I only see the Validation steps using the UCI Machine Learning data. Where is this? If not used, how can you scientifically justify skipping this?

Please answer the above eight questions.

Reviewer 2 Report

Please see the attached report

Round 2

Reviewer 1 Report

I am satisfied with the responses to my questions. I still feel that better verification & validation (V&V) could be done, but for the time being it is acceptable.